# The Action Cycle Theory of Perception and Mental Imagery

**David F. Marks** [ID]

Independent Researcher, Provence-Alpes-Côte d'Azur, 13200 Arles, France; dfmarksphd@gmail.com

**Abstract:** The Action Cycle Theory (ACT) is an enactive theory of the perception and a mental imagery system that is comprised of six modules: Schemata, Objects, Actions, Affect, Goals and Others' Behavior. The evidence supporting these six connected modules is reviewed in light of research on mental imagery vividness. The six modules and their interconnections receive empirical support from a wide range of studies. All six modules of perception and mental imagery are influenced by individual differences in vividness. Real-world applications of ACT show interesting potential to improve human wellbeing in both healthy people and patients. Mental imagery can be applied in creative ways to make new collective goals and actions for change that are necessary to maximize the future prospects of the planet.

**Keywords:** Action Cycle Theory; perception; mental imagery; vividness; VVIQ; affect; schemata; action; individual differences; neuroscience

## 1. Historical Context

Since the time of Descartes, perception has been conceived as the reception of sensory information that may or may not lead to any behavioral response or action (Figure 1).

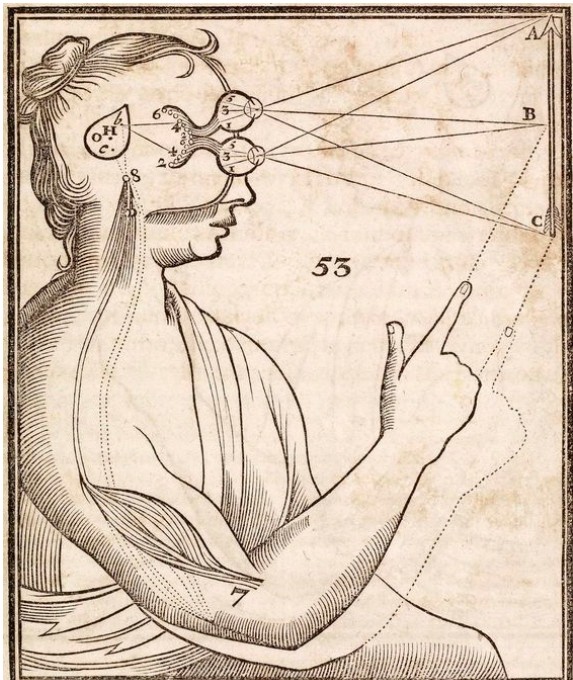

**Figure 1.** Descartes' 1644 illustration from *Principles of Philosophy* [1] showing his theory of vision. Light rays from the arrow stimulus (A,B,C) impress particles into the eyes from which the image is transmitted to the pineal gland, the confluence of mind and body. It is as if the visual system is an internal screen or theatre [2]. The picture shows the pineal gland converting an external stimulus into the action of pointing.

Except for the removal of the pineal gland from this system, this 'S-R' or theatre model of perception as passive stimulation (S) of an empty organism to produce a response (R) persisted for almost three centuries until Hermann von Helmholtz (1910/1925) ([3], p. 27) offered a more nuanced approach with his proposal of 'unconscious inference'. Helmholtz held that unconsciously formed 'inductive conclusions' led to the formation of sense-perceptions. With notable exceptions, the psychology of perception in the first half of twentieth century was dominated by the S-R framework of behaviorism in the US and Pavlovian conditioning in Russia. In his 2001 Medawar Lecture, Richard L. Gregory stated:

> "Yet, the most striking fact is that perceptual experience is far richer than available retinal images; and though neural signaling is slow, it is not usually delayed in time. From these shadowy ghosts in our eyes we see hard solid objects with properties beyond optics. This depends on knowledge of objects, and how they interact, allowing behavior to be appropriate to what is known or assumed, rather than limited to what is being sensed. This is where knowledge comes in, as the past enriches the present, and allows some prediction into the future" [4].

Seldom, if ever, can vision provide an exact copy of sensory stimuli, because both 'top-down' and 'bottom-up' processes are involved. The empty organism was filled by Jerome S. Bruner and Leo Postman in 1947 with the suggestion that 'top-down' processes of goals, expectations, and prior knowledge acted as top-down influences on perception [5]. Hubel and Wiesel (1962) [6] discovered that the visual presentation of stimuli with oriented lines lead to characteristic firing patterns in the primary visual cortex, suggesting the existence of 'bottom-up' visual cue detectors. The so-called 'Cognitive Revolution' of the 1950s brought new theories about top-down processing, and Ulric Neisser's (1967) book *Cognitive Psychology* [7] presented an innovative theory of the 'perceptual cycle' as a constructive process for anticipating objects and events. Although Neisser's theory appeared flawed, it sowed the seeds for the Action Cycle Theory (ACT) of perception and mental imagery [8,9]. The three theories of VMI that existed in 1990 when the ACT was proposed were reviewed by Thomas [10]. The approach of Jeannerod and Decety (1995) [11] shows similarities to ACT, and there is also symbiosis with the approach of O'Regan and Noë (2001) [12], yet both of these approaches neglected emotion. Before continuing, a few definitions and assumptions need to be specified.

Visual perceptual imagery (VPI) is a subjective experience that occurs when there is stimulation of the retinae with light.

Visual mental imagery (VMI) is a subjective quasi-perceptual experience that occurs in the absence of the relevant object, with or without stimulation of the retinae with light.

Mental imagery can occur in any sensorimotor modality and in a multitude of states of waking and sleep that range between voluntary and conscious to automatic and unconscious.

A key concept is 'vividness', which is a combination of clarity and liveliness. The more vivid an image, the closer it approximates an actual percept.

VMI is activated by a set of neurological processes that are also involved in visual perceptual imagery (VPI). Research relevant to the Action Cycle Theory has utilized the *Vividness of Visual Imagery Questionnaire* (VVIQ) [13,14] and other similar instruments to investigate the influence of imagery vividness on the six core processes. As research using the VVIQ provides the foundation for the evidence to be presented here, it is helpful to indicate the VVIQ's credentials by summarizing a few issues that have been raised about its use as a measure of imagery vividness. The VVIQ is not the only available instrument for the measurement of imagery vividness, and it needs to be noted that mental imagery is both visual and multisensory in nature. For multisensory imagery research, a suitable measure is the *Plymouth Sensory Imagery Questionnaire* [15]. Vividness questionnaires are also available for research in different sensory modalities and languages, including the *Vividness of Movement Imagery Questionnaire* (VMIQ) [16], and in special domains such as the *Vividness of Wine Imagery Questionnaire* [17].

Vividness is found to benefit imagining, remembering, thinking, predicting, planning, and acting. Originally published in 1973, the VVIQ enabled an empirical approach to the measurement of VMI vividness [13,14]. Multiple investigators have conducted studies using the VVIQ, VMIQ [16] and VVIQ-2 [18]. Alfredo Campos and María José Pérez-Fabello [19] reported high estimates of internal consistency, reliability and construct validity for the VVIQ and VVIQ-2. Although a component of social desirability occurs in VVIQ scores [20], as for any questionnaire, the scores are sufficiently robust to predict a broad range of brain states, psychophysiological variables and other objective indicators. A key methodological study by Matthew Runge et al. used meta-analysis to compare VVIQ scores with trial-by-trial vividness ratings against objective criteria to address predictive validity [21]. Trial-by-trial vividness reports produced significantly larger effect sizes than VVIQ across three experimental methodologies, with neural measures yielding significantly greater effect sizes than behavioral and cognitive ones. The majority of findings reviewed here used VVIQ scores and not trial-by-trial vividness reports, and so, most likely, they underestimate the strength of associations between VMI vividness and the key variables presented in the sections below.

The construct of 'vividness' is intrinsically subjective, but vividness measures can provide clues about the nature and function of mental imagery in studies that compare groups or individuals with high and low values along the vividness continuum. Assessments of vividness using validated instruments provide a fruitful evidence base that is relevant for ACT. Controlled laboratory investigations with objective tasks evaluate perceptual modelling, recognition, recall, other cognitive performances, behavior, clinical cases and electrophysiological techniques such as BOLD functional magnetic resonance imaging (fMRI), transcranial electrical stimulation (TES) and near-infrared spectroscopy (NIRS) to map anatomical and functional brain connectivity, for example [22,23]. Vividness measures significantly correlate with the degree of similarity between VMI and VPI neural responses for actual objects and physical activities [24–28].

A landmark meta-analysis of 150 studies published by Stuart J McKelvie in 1995 was able to demonstrate high levels of reliability, content validity and criterion validity of the VVIQ [29,30]. McKelvie's meta-analysis showed that the vividness of conscious mental imagery is strongly associated with the performances that are most likely to benefit from perceptual-motor imagery and mental practice.

## 2. Action Cycle Theory

Action Cycle Theory (ACT) is an enactive theory of perception and mental imagery with six interrelated modules (Figure 2). According to the theory, the six modules are viewed as necessary and sufficient for perception and mental imagery to occur. Figure 2 summarizes the theory in showing that perception and mental imagery are constituted by the six modules, which are: *Object*, *Schemata*, *Action*, *Affect*, *Goals* and *Other's Actions*. The arrows indicate hypothesized causal relationships. 'Action' refers to the perceiver's/imager's own actions. 'Others' behavior' refers to the observed actions of others.

In ACT, the perceptual system involves an exploratory cycle that harmonizes interactions with objects motivated by the drive to maximize equilibrium and positive affect. Schemata of sensorimotor 'templates' or 'mental models' detect cues and minimize threats (type-S schemata) and generate actions (type-M schemata). The ACT is supported by an extensive body of evidence concerning the cognitive, affective and neurophysiological aspects of mental imagery.

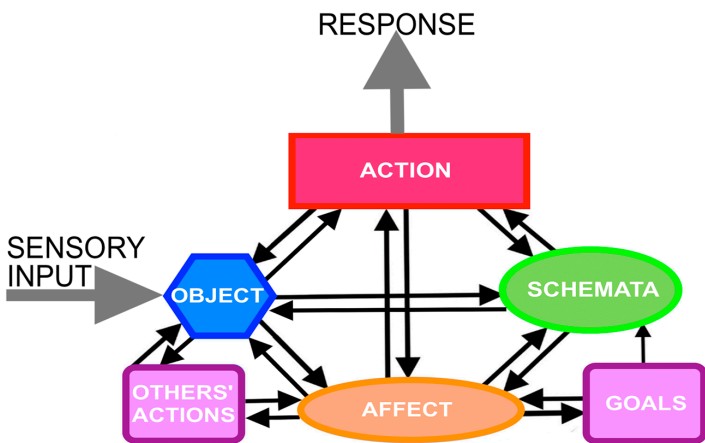

**Figure 2.** The six modules of Action Cycle Theory: *Object*, *Schemata*, *Action*, *Affect*, *Goals* and *Other's Actions*. The large grey arrows represent sensory input and the organism's response in perception. In mental imagery, the system of six modules is activated in the absence of sensory input. The black arrows indicate hypothesized causal relationships. The majority of interconnections are reciprocal, meaning that any module can causally influence, and be causally influenced by, its neighbors. The absence of an arrow going from Schemata to Goals indicates that Schemata do not influence Goals that are determined only by Affect, i.e., the degree to which happiness or pleasure can be increased and/or pain can be diminished.

A recent monograph [31] placed the ACT within the General Theory of Behavior. Accordingly, motivation is held to be universal pro-social striving towards security, stability and equilibrium driving organisms to adapt to the environment and to one another. The General Theory holds that the primary motivation of natural organisms is a psychological form of homeostasis for behavior that acts in parallel with physiological homeostasis in the regulation of bodily systems. Acknowledging that self-control of behavior mostly proceeds unconsciously and automatically, there is also a need for an executive control system for voluntary actions. That executive function is served by consciousness, which is also an ipseity center for the integration and regulation of motivation with feelings and actions by the self.

The author's description of mental imagery as an enactive, quasi-perceptual process is stated as follows:

A "life-like" image allows a person to experience it in a very real way. "Evoke" means to cause someone to sense or feel something. I appreciate this fact from personal experience. When I am watching a film of people lighting a campfire, or smoking a cigarette, I actually can "smell" the smoke at the campfire or the stench of a cigarette. These olfactory imaginings occur as if I am actually carrying out the activity, albeit on a lesser scale. Note that it is an activity, not a picture. Evoking a vivid image produces a life-like activity of "seeing", "hearing", "tasting", "smelling", "touching" or "feeling" something; mental imagery is a sensory-affective process that resembles, but is not identical to, perception with action [32].

A persistent philosophical objection to the use of self-reported imagery vividness ratings in the VVIQ and similar instruments relates to their subjective nature. Critics suggest that the impossibility of a shared reference point means that individuals are unable to reliably rate their imagery vividness along a vividness scale [33,34], and so vividness ratings are 'too' subjective for studies of behavioral and brain states. As plausible as these criticisms may seem a priori, a multitude of controlled studies demonstrate that VVIQ scores are reliable, valid and sufficiently robust to make accurate predictions that, more frequently than not, are verified by objective indicators. The challenge of the subjectivity issue was addressed from the beginning of VVIQ research when the author used objective tests of picture recall to validate people's VVIQ vividness scores [13,14], which, curiously, critics ignore. Between one and two minutes after the presentation of colored pictorial scenes, a

recall task required participants to answer questions about the position, color and detail of items within the stimulus array. Groups of people who reported high vividness produced higher accuracy of recall than groups who reported low vividness. The significant association between recall accuracy and VVIQ vividness scores was replicated in three different experiments, suggesting a robust finding. Dozens of investigators independently reported validating studies using a variety of objective performance indicators to compare high and low vividness groups [11]. Multiple neuropsychological and neuroimaging studies have since demonstrated that the VVIQ provides a valid measure of VMI vividness [18–23]. Illustrative examples are presented in the following sections, indicating strong empirical evidence to support the six facets of ACT, starting with schemata.

### 3. Schemata

Schemata are internal models that reflect commonalities across multiple experiences of objects and actions [35–37] while enabling the body to maintain its composure [38,39]. Rumelhart [35] states: "Schemata are employed in the process of interpreting sensory data, in retrieving information from memory, in organizing actions, in determining goals and subgoals, in allocating resources, and, generally, in guiding the flow of processing in the system. Perhaps the central function of schemata is in the construction of an interpretation of an event, object, or situation—that is, in the process of comprehension". The body schema was invoked in neurology by Head [38] and in psychology by Piaget [39], Bartlett (1932) [40] and others [41,42]. Schemata are thought to be of three kinds: (i) scene schemata (S-type) for observing, identifying and completing spatial patterns in scenes; (ii) motor schemata (M-type) for observing, mirroring, synchronizing, planning, generating and predicting actions and (iii) body schemata (B-type) for controlling and maintaining posture, gait and balance. All three operate automatically, unconsciously and quickly. Schemata are adaptive and readily accommodating to the flux of a continuously changing environment. Schemata act as mental models or templates for rapid processing within perceptual-motor systems including posture, balance and gait. Schemata are constructed and controlled across the hippocampal region, cerebral cortex and cerebellum, which play an extensive role in mental activities [43], as indicated in Figure 3.

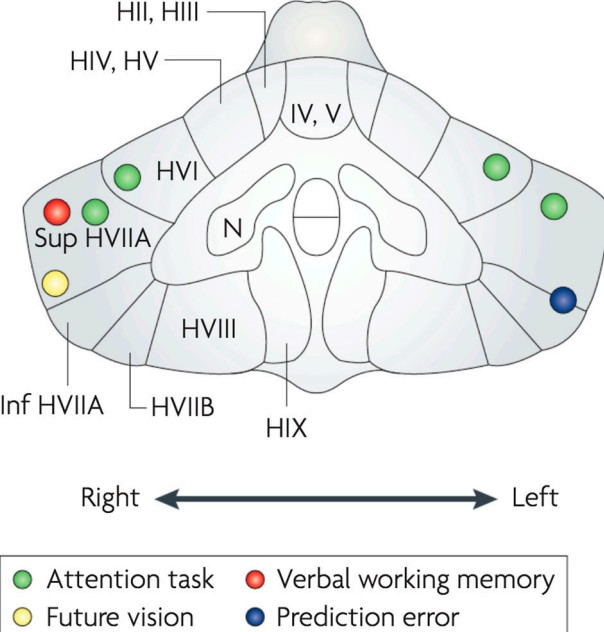

**Figure 3.** Mental activities controlled by schemata in the cerebellum. The figure shows a coronal section of a human cerebellum on which the sites of the observed activities are indicated by colored circles. Reprinted with permission from ref. [43]. 2008, M. Ito.

The cerebellum transmits information through the brain stem, via the thalamus, to the hippocampus, to other regions of the temporal lobe and to other regions of the cerebral neo-cortex (including parietal regions) [44–49]. Parkins [50] suggests that the output from the cerebellum can be perceived as action images (a felt sense of doing), visual images or auditory images: "These mental images may be experienced as daydreams, spontaneous thoughts, internal speech, or an inner voice. They may be experienced at various levels of intensity from faint imaginings to vivid hallucinations" [50].

Partly through its strong bidirectional connections with the cerebellum, it is evident that the hippocampus is another key structure for imagery schemata. Recent studies have demonstrated that imagery VVIQ vividness scores correlate significantly with objectively quantifiable measurements of hippocampal volumes in a large sample of healthy people across different ages [51]. Tullo et al. (2022) investigated the interactions among scene-selective brain regions including the parahipoccampal place area (PPA), the retrosplenial complex and the occipital place area (OPA) using Dynamic Causal Modelling for resting-state functional magnetic resonance data [52]. They tested whether resting-state effective connectivity among scene-selective regions reflected individual differences in VMI vividness using the VVIQ. An inhibitory influence of occipito-medial on temporal regions was observed, as well as an excitatory influence of more anterior, medial and posterior brain regions. The connection strength from OPA to PPA was especially high in the left hemisphere, with the signal between these regions showing positive correlations with vivid mental imagery ability (Figure 4).

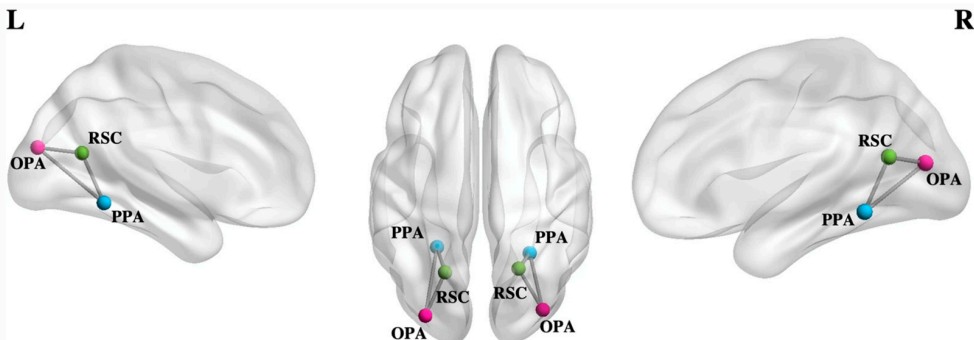

**Figure 4.** Anatomical location of regions of interest. Nodes of each region are displayed in different colors: the parahippocampal place area (PPA) is represented in sky-blue; retro-splenial complex (RSC) is represented in green; and occipital place area (OPA) is shown in pink. The edges between regions represent the connections separately modelled for each hemisphere in the Dynamic Causal Modelling (DCM) analysis. Reprinted with permission from ref. [52]. 2022. M G Tullo. Regions of each hemisphere were visualized using the BrainNet Viewer [53].

Several different studies confirm that a network of brain areas support VMI, spatial cognition and navigation, including the right hippocampus and right caudate nucleus that are associated with knowing where places are and navigating accurately to and between them [54]. There appears to be in-built redundancy in the VMI network, as evidenced by the reported ability to experience vivid VMI in the absence of significant amounts of cortical tissue (depending, of course, on the amount and location of the missing or damaged tissue). In spite of near-complete cortical blindness, one person exhibited vivid visual imagery. The pattern of cortical activation was found using fMRI to be indistinguishable from that of sighted subjects, in contrast to the visual perceptual responses, which were greatly reduced [55]. These findings underscore the role of the cerebellum and hippocampus in VMI that can remain vivid even with missing cortical tissue.

To date, no consensus has been established about the cortical systems involved in VMI. One large-scale meta-analysis of 46 fMRI studies [56] revealed that VMI engages fronto-parietal networks and a region in the left fusiform gyrus. Spagna et al. [56] proposed a revised neural model of VMI with fronto-parietal networks initiating, modulating and

maintaining activity in a core temporal network centered on the fusiform imagery node, a high-level visual region in the left fusiform gyrus. The cortical basis of VMI remains a subject of active investigation. I turn now to consider perception and mental imagery of objects.

## 4. Objects

Perception and mental imagery of objects requires a network of structures at different levels of the CNS. The earliest investigations focused on the occipital cortex, yet many key studies have demonstrated that VMI and VPI share a diverse set of structures across interconnected regions including the brain stem and the cerebellum. Cui et al. (2007) asked participants to imagine a visual scene, such as an ant crawling on a checkered tablecloth toward a jar of jelly [25]. The investigators found that vividness correlated with early visual cortex activity relative to the whole brain activity measured by fMRI and also performance on a psychophysical task (Figure 5). The results showed that individual differences in the vividness of mental imagery are objectively quantifiable.

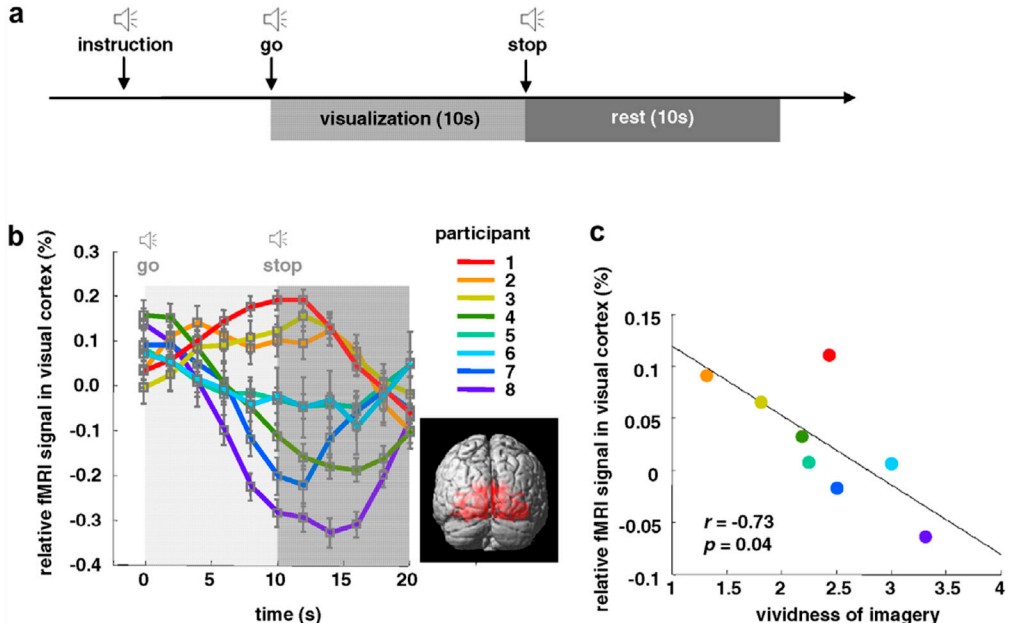

**Figure 5.** Relative activity in visual cortex correlates with subjective vividness rating. (**a**) Timeline of the visualization task. Participants began to visualize upon hearing the 'go' signal and stopped visualization upon hearing the 'stop' signal, resulting in a 10-s visualization phase and 10-s rest phase. All instructions were auditory. (**b**) Time course of the relative fMRI signal in visual cortex for 8 participants. Relative fMRI was taken as the BOLD signal in early visual cortex (Brodmann's areas 17 and 18, illustrated in inset) minus the BOLD signal measured over all of grey matter. For plotting purposes, participants are ordered by their relative visual cortex activity averaged over the visualization window of 0–10 s. The negative signal for some subjects is due in part to the subtraction of the whole brain activity—i.e., other regions can increase more than the visual cortex during the time window. (**c**) The relative visual cortex signal averaged over the visualization window correlates significantly with the subjective rating of vividness ($p = 0.04$). Reprinted with permission from ref. [25]. 2005. X. Cui et al. Note that there were only 8 participants in this study, and the $p$ level for the correlation was only 0.04. The 5 subjects with the most vivid visual imagery produced positive relative fMRI signals in the visual cortex. Other studies have obtained different results [27].

The action cycle attempts to maximize imagery vividness by controlling eye movements, so that vivid images are associated with fewer eye movements [57,58]. It is likely that keeping the eyes still provides the benefit of avoiding new perceptual input. This hypothesis is consistent with Amedi et al.'s [59] findings that people 'switch off' the auditory

modality, with auditory cortex deactivation negatively correlating with activation in the visual cortex and with the score in the subjective vividness of visual imagery questionnaire (VVIQ). The action cycle optimizes vividness by automatically attenuating distracting sensory channels (i.e., auditory, somatosensory and subcortical visual structures). Amedi et al. (2005) [59] compared VMI and VPI directly using visual objects as stimuli. VMI but not VPI was associated with *deactivation* of the auditory cortex as measured by BOLD fMRI. During VMI, auditory cortex deactivation was found to negatively correlate with activation in the visual cortex and with VVIQ scores (Figure 6). The authors suggested that pure visual imagery corresponds to the isolated activation of visual cortical areas with the concurrent deactivation of "irrelevant" sensory processing that could disrupt the image created by our "mind's eye" [59].

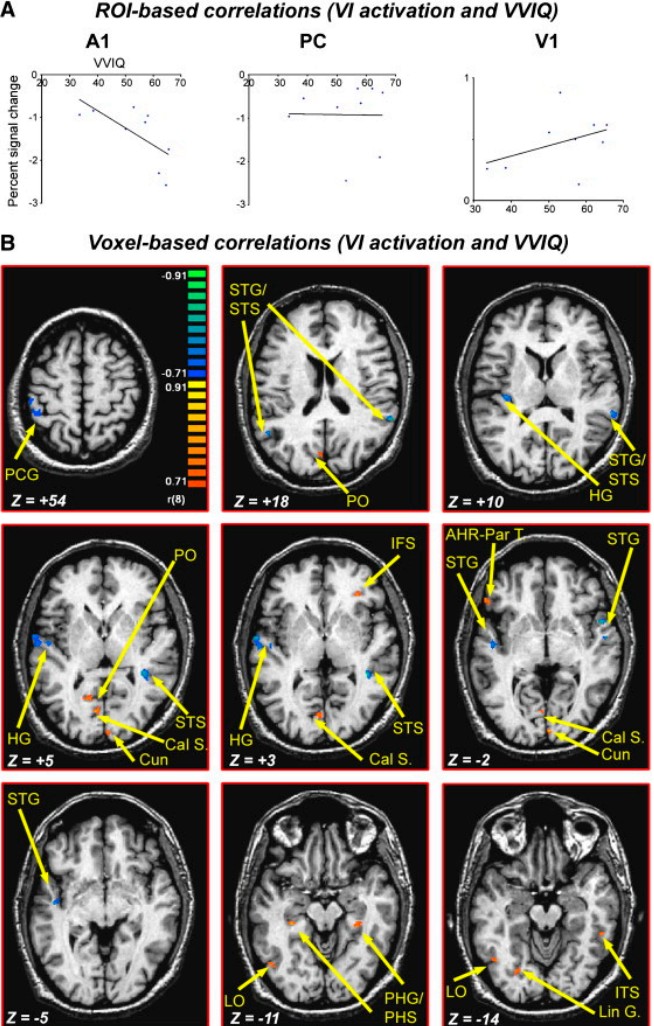

**Figure 6.** Correlation between VVIQ scores and VI BOLD signal. (**A**) Correlations between the percent signal change for VI in each subject and the VVIQ scores of these individual subjects in three ROIs. There were significant negative correlations between A1 (primary auditory cortex) activity and the vividness of the imagery, a trend for positive correlations with V1 (primary visual cortex) and no correlation with PC (posterior cingulate 'default brain' area). (**B**) Voxel-by-voxel correlations between VI activity and individual VVIQ correlations. Results presented on axial slices covering mainly auditory cortex. The investigators found negative correlations in auditory and somatosensory cortex only. Positive correlations were found in visual, prefrontal and parahippocampal areas bilaterally. Reprinted with permission from ref. [59]. 2005. A. Amedi et al.

Perceptual images of objects require the retinal blind spot and occluded objects to be filled in and completed. Age-related eye diseases such as microdegeneration, myopia and refractive errors, cataract, diabetic retinopathy and glaucoma, amblyopia and strabismus also limit the quality of perception. According to Helmholtz, if a spot or blemish occurs on an image, it will be filled in: *"the spectator will at once supply the gap by his imagination and fill it up with the colour of the background"* ([7], p. 209). The retinal blind spot contains no photoreceptors, and missing perceptual data must be automatically filled in, so that normally, one does not perceive any gaps. Revina and Maus (2020) [60] compared perceptual filling-in of the blind spot with that occurring in the case of artificial gaps of identical size and eccentricity. They found stronger perceptual filling-in occurred in the blind spot than for artificial gaps, especially in the case of vivid imagers.

Amodal completion is a normal process of perception that can be taken for granted, yet the precise mechanism remains controversial [61]. The invisible backside of any solid object that is 'self-occluded' is also completed by a similar process. One of the main questions is whether the filling-in is achieved by mental imagination. Three possible mechanisms for amodal completion are discussed by Nanay (2018) [62]: (i) amodal completion is a type of perception similar to sensory stimulation-driven perception; (ii) amodal completion is achieved by having beliefs; (iii) amodal completion occurs using mental imagery. Nanay argues that a body of empirical findings support (iii) but not (i) or (ii). In line with ACT, a fourth explanation is that special type-S schemata are deployed to trigger exploratory actions, actual or imagined. Proximal objects can be reached, grasped, manipulated, viewed from a different angle, studied with head movements or by moving and circling around an object to confirm its nature and identity. Thus, amodal completion consists of enactive imagistic exploration.

Vision under poor lighting, noise or camouflaging requires top-down processes to enhance selectivity and perceptual content that can sometimes cause anomalous experiences or even illusions. Investigating face detection in pure noise images using fMRI, Zhang et al. [63] found greater activation for face versus nonface responses in the fusiform face area, but not in the occipital face area. Medial frontal, parietal, supplementary motor, parahippocampal and striatal areas produced negative correlations between face-detection activation and behavioral responses indicating top-down influences. Salge et al. [64] extended these findings by showing that vivid imagers were more likely to see faces in noise, and the boundary between imagination and 'reality' appeared narrower in vivid imagers who were also more prone to anomalous experiences in noisy sensory environments. I turn next to action.

## 5. Action

We began this article with a Cartesian 'theatre' model of vision with an inner screen. O'Regan (1992) [2] argued against this straw man as follows: *"the outside world is considered as a kind of external memory store. This can be accessed by casting one's eyes to some location. The feeling of the presence and extreme richness of the visual world is, under this view, a kind of illusion created by the immediate availability of the information in this external store".* In line with ACT, this account consists of getting to know or verifying the sensations caused by easily made exploratory actions, real or imagined. In O'Regan's account, as in ACT, perception is entirely enactive [2] (p. 472). Others make a similar point [65–67]. Contra O'Regan, however, according to ACT, every image includes one or more schema of each type, that is, type-S, type-M and type-P schemata [68,69]. One looks, one sees, one moves one's eyes in unceasing cycles with other body parts participating, especially the head.

The association between vivid imagery and covert psychomotor activity has been acknowledged since William Shaw published "The relation of muscular action potentials to imaginal weight-lifting" in 1940 [70]. Shaw had discovered action potentials showing muscular activity during imaginal lifting of a weight. As in actual weight lifting, during imagining, the action potentials increased linearly with the magnitude of the weight, and it was more prominent with vivid imagery. Mental imagery influences nascent muscular

activity, and vice versa. As a corollary of this, there should also be a measurable influence of physical actions on nascent mental imagery in which congruent action enhances image vividness. This prediction was confirmed by Shinsuke Hishitani, Takuya Miyazaki and Hiroki Motoyama, who demonstrated that imagery vividness is enhanced by moving an index finger with one's eyes closed as if drawing the contents of the mental image [71].

Vividness differences are found across tasks and specialist groups with higher or lower than average skill levels, e.g., air traffic controllers, pilots, Olympic athletes [72], orchestra conductors [73] and surgeons [74]. Research carried out by Anne Isaac and the author included a sample of children aged 7–15 years with developmental difficulties in movement control. This sample of children was found to be extremely poor imagers, with 42 per cent reporting no imagery at all [72]. In the same study, physical education students reported more vivid imagery than students specializing in physics, English and surveying, and significant differences were found between elite athletes' imagery and that of matched controls. Air traffic controllers and pilots were found to have significantly more vivid imagery than matched controls. Self-reports of imagery vividness show a systematic pattern of relationships with age, gender and specialization requiring high-level performance of perceptual motor skills. These findings highlight the connections between mental imagery and the observation, practice and implementation of actions, which can be made all the easier by specialist neurons that mirror the actions observed in others.

The 'mirror neuron' that is responsive to the observation of motor actions by others was originally identified in the macaque monkey [75]. Then, a similar human mirror neuron system was found containing cortical neurons that respond to observation of other individual's actions [76] and one's own physical movements [77]. This specific form of motor enactment suggests that type-A schemata are able to create or re-create motor repertoires that are transferrable between individuals, enabling motor regions of the brain to be retrained simply by observation [78,79]. I return to the role of others' actions in Section 8 below.

Enactive mental imagery is utilized in the development of brain–computer interfaces (BCI), which have brain activity as input and an action carried out by an external device as output. BCI enable humans to interact with the environment without using their muscular system [80] by exploiting motor imagery of users who imagine moving a body part to cause a change in the activity of the motor cortex (MI-BCI) [81]. The MI-BCI system learns to classify brain activity changes and carry out commands that can be of benefit to people with motor impairments [82] following stroke or aging [82]. An investigation of the influence of visual imagery vividness on MI-BCI used VVIQ scores of novice BCI-users in a left- versus right-hand motor imagery task [83]. People with vivid imagery were able to produce more skilled MI-BCI performances.

Blumen and colleagues (2014) examined the behavioral and neural correlates of imagined walking (iW), imagined talking (iT) and imagined walking while talking (iWWT) [83]. Thirty-three 'cognitively-healthy' older adults of an average age 73 years performed iW, iT and iWWT during fMRI. A pattern of brain regions was found that: (1) varied with imagery task difficulty; (2) involved cerebellar, precuneus, supplementary motor and other prefrontal regions; and (3) was associated with kinesthetic imagery ratings and behavioral performance during actual WWT. I turn next to consider affect.

## 6. Affect

Since Freud's famous analysis of Anna O., clinical case studies of the imagery–emotion link remain fascinating but difficult to interpret. The links between imagery and emotion have been said to involve a 'special relationship' comprising imagery's direct influence on emotional systems in the brain to sensory signals [38]. However, research on emotion tends to be piecemeal and lacking in cohesion. For ethical and practical reasons, 'hot emotion' is not an appropriate topic for laboratory methods except with the most anodyne stimuli in the form of words, symbols, patterns and pictures. More complex stimuli such as narratives, stories, music and films have greater ecological validity, but the difficulty

separating cause from effect makes their use problematic. Findings of meta-analysis of fMRI studies of autobiographical memory, to give one example, indicate a complex array of primary, secondary and tertiary brain regions that appear messy and hard to interpret [84]. In spite of the difficulties, innovative investigations have been making progress.

Again, one focus has been on visual imagery vividness, which appears to play a role in perceptual decision making following threatening cues. It enhances the planning of actions to protect and to maximize safety, stability and security in light of goals, drives and others' behavior. The bio-informational theory of emotional imagery of Peter J. Lang [85] has been influential. One study used a task in which participants saw threat-related and neutral cues while detecting perceptually degraded fearful and neutral faces at predetermined perceptual thresholds [86]. Threat cues were found to improve the accuracy, sensitivity and speed of perceptual decisions compared to neutral cues. Vivid imagery ability appeared to enable more vivid threat-related type-S schemata, facilitating subsequent perception. After controlling for performance related to neutral cues, high vividness VVIQ-2 scores were associated with more precise and rapid decision making following threatening cues.

Köchel et al. (2011) [87] used multi-channel near-infrared spectroscopy (NIRS) with 35 healthy adults to study their cortical responses to disgusting, happy and neutral pictures and their associated VMI. Their VPI provoked increased oxygenated hemoglobin in occipital regions, whereas VMI was associated with an increase in parietal areas. Affective pictures (disgusting and happy) provoked greater activation compared to neutral ones, especially in the left occipital cortex, during both VPI and VMI. Participants with higher vividness scores on the VVIQ showed more occipital activation during affective imagery, suggesting occipital activation may be necessary for vivid VMI.

Wicken, Keogh and Pearson (2021) [88] examined the skin conductance of aphantasics (people who claim no awareness of VMI) with a control group while reading and imagining frightening stories. The results indicated a flat-line physiological response to reading and imagining frightening stories but not when perceptually viewing fearful images when the two groups reacted similarly (Figure 7). These findings support the ACT in showing that strong affect is causally related to vivid VMI.

The powerful causal links between emotion and imagery have been exploited in multiple mental health and neurological interventions and to minimize distress in patients entering hospital for procedures such as surgery. In a controlled trial with patients receiving abdominal surgery, we tested the effects of guided imagery as a preoperative preparation to increase patients' feelings of coping with surgical stress [89]. Twenty-six coping imagery patients were compared with 25 controls who received no imagery training but only background information about the hospital. We found that state-anxiety was similar in the two groups prior to the intervention, but the guided imagery patients experienced less postoperative pain than the controls, were less distressed by it, felt that they coped with it better and requested less analgesia. Cortisol levels measured in peripheral venous blood did not differ on the afternoon of admission before preparation but were lower in imagery patients than in controls immediately before and after surgery. Noradrenaline levels were greater on these occasions in imagery patients than controls. Two predictions from ACT are confirmed by the results of this study: (i) VMI produces emotional arousal; (ii) the rehearsal of coping with pain ameliorates the adverse emotional arousal associated with pain. Surgeons themselves receive a benefit from mental practice using VMI to reduce their levels of emotional stress [90,91]. I turn next to goals.

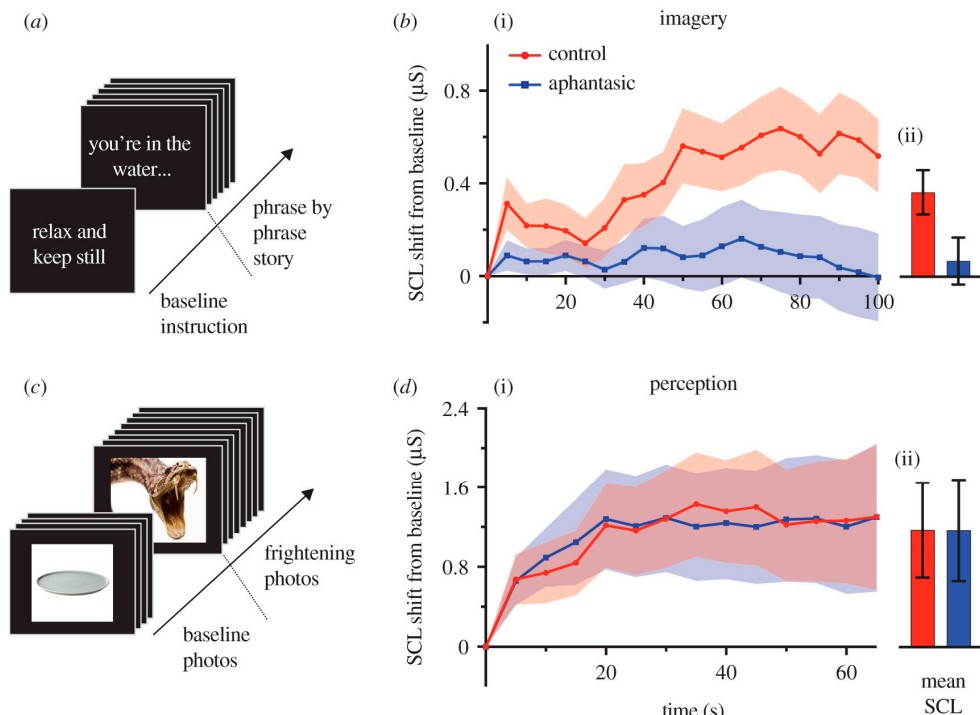

**Figure 7.** Skin conductance data for imagery and perception experiments with aphantasics and controls. (**a**) Imagery experiment. Fifty seconds of baseline SCL was recorded prior to each scenario trial while participants viewed an on-screen instruction. Next, each scenario trial was presented to participants as a succession of 50 on-screen phrases, each displayed for 2 s. (**b**) (i) Aggregated progressions of baseline-corrected SCL across the duration of scenarios (sampled as average across 5 s time bins). (ii) Mean and SEM across time bins. (**c**) Perception experiment. Baseline SCL was recorded while participants viewed neutral photos, before being presented with a succession of frightening photos. Photos appeared on screen for 5 s each and immediately followed one another. (**d**) (i) Aggregated progressions of baseline-corrected SCL across the duration of the frightening photos sequence (sampled as average across 5 s time bins). (ii) Mean and SEM across time bins. Reprinted with permission from ref. [88]. 2021. M. Wicken et al.

## 7. Goals

Mental images are strongly linked to goals, including personal, family and community goals. Images reflect specific goal content and act to maintain goals. A primary function of mental imagery is the selection, rehearsal and planning of goal-directed activity. ACT assumes that mental rehearsal enables the person to guide and improve actions in a way that would be more difficult without it. It is assumed that an action goal includes an internal representation of the external object toward which the action is directed and toward the final state when the object is reached. Conscious mental imagery provides the primary mechanism for goal-directed action planning that needs to be adapted to novel circumstances, whereas unconscious goal-directed action can occur for automatic, habitual actions such as walking or typing.

ACT proposes reciprocal causation between imagery, goals, emotion, action, others' action and schemata. New situations require new images that lead to new goals, and vice versa [92]. Visualizing goals requires the imagination of possible future scenarios and selves. Vivid visualizers compared to individuals with less vivid imagery are able to construct events and induce more intense emotions and motivation. Motivation to change and to facilitate change can be raised by vividly imagining the change that one desires. In one study, it was found that high vividness VVIQ scores produced more visual details, other sensory details, clarity of the spatial context and feeling of emotions, and the intensity and personal importance of future goal-related events was stronger [93].

The different perspectives one uses in imagery of the self can make a difference to future action. There is the visual perspective of oneself from the first-person perspective versus taking an observer's third-person perspective. One study took place on the eve of the 2004 U.S. presidential election when registered voters in Ohio were instructed to use either the first-person or the third-person perspective to mentally image themselves voting in the election [94]. Imaging voting from the third-person perspective caused participants to adopt a stronger pro-voting mindset. This effect carried over to behavior, causing those adopting the third-person perspective to be significantly more likely to vote in the election.

The motivational power of mental imagery is strengthened through its close link with emotions. Boomsma, Pahl and Andrade [95] used an *Environmental Mental Imagery* scale to assess how vividly and frequently participants could imagine problems related to the goal of reducing plastic debris floating in the oceans after seeing a video. There were moderately strong correlations between the participants' environmental imagery vividness and their pro-environmental goals (r = 0.51, $p < 0.001$). Emotive environmental imagery can be internalized as mental images to influence individuals' personal pro-environmental goals and mitigation strategies.

Innovative imagery approaches are available to improve public engagement with climate change [96]. Elsewhere, this author has suggested that, in line with ACT, global problems caused by human behavior such as warming, poverty and overpopulation may be helped by applying mental imagery in creative ways [97]. Human and planetary survival may ultimately depend on the effective use of the human imagination and especially enactive mental imagery. In designing solutions to global problems such as global warming and population growth, there is no machinery to rival the human imagination when motivated by strong pro-environmental goals. Thus, motivation is the key. Finally, I turn to the role of others' behavior.

## 8. Others' Behavior

Perception is an enactive visuomotor exploration by the eye, hand and body of objects, scenes and others' actions. As previously noted, the discovery of mirror neurons was a significant step toward an increased understanding of perception [98]. Mirror neurons participate whenever other's actions are being observed. They are not a necessity for self-generated action, unless one is basing an action on a mental image of another actor.

Mirror neurons fire not only when a person is performing an action but when passively observing a similar action performed by another. Fully consistent with ACT, mirror neurons are designed for prosocial functions of action and emotional understanding or empathy, synchronization and psychological homeostasis. There appears to be a specific kind of type-M schemata for mapping others' actions, emotions and communications. A meta-analysis of 125 fMRI studies revealed an extensive brain network with mirror-like properties [99]. These regions included a core fronto-parietal network that is active during the observation and execution of actions, with additional mirror areas recruited during tasks that engage non-motor functions.

Studies using BOLD fMRI have mapped cortical activation during reaching, observed reaching and imagined reaching in humans [100]. A strong overlap was found between executed, observed and imagined reaching activations in the dorsal premotor cortex as well as in the superior parietal lobe and the intraparietal sulcus, in accordance with ACT. The mirror neuron system is specific to the type of hand action that is performed (Figure 8).

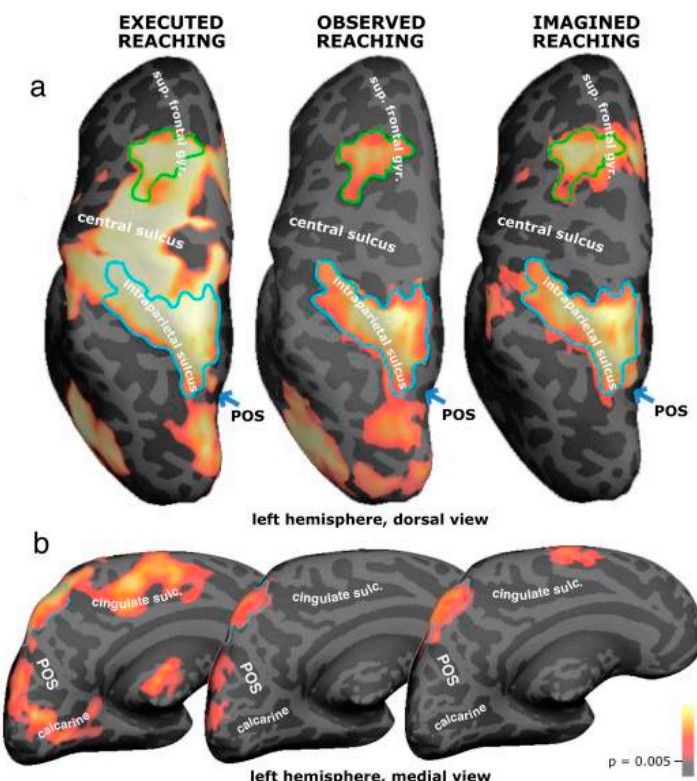

**Figure 8.** Overlap between executed, observed and imagined reaching in left dorsal premotor (superior frontal sulcus and gyrus) and left posterior parietal areas, on group surface-averaged activations from 15 subjects, displayed on one subject's inflated hemisphere. The overlaps in premotor and parietal regions served as regions of interest in the percent signal change analysis. (**a**) Dorsal view of left hemisphere. (**b**) Medial view of left hemisphere. Executed, observed and imagined reaching all activated a medial parietal area located in between the parieto-occipital sulcus and the posterior end of the cingulate sulcus, outlined in light blue. Sup. frontal gyr. = superior frontal gyrus; POS = parieto-occipital sulcus; calcarine = calcarine sulcus; cingulate sulc. = cingulate sulcus. Reprinted with permission from ref. [100]. 2007. F. Filimon, et al.

Similar to mirror regions for observing others' actions, emotion mirror regions activate neural schemata for particular emotions. The empathic emotion of disgust was noted by Charles Darwin: "Disgust refers to something revolting, primarily in relation to the sense of taste, as actually perceived or vividly imagined; and secondarily to anything which causes a similar feeling, through the sense of smell, touch and even eyesight" (Darwin, 1872/1965) [101]. Humans and other animals have the capacity to perceive and to understand emotional responses in others. The same neural mechanisms underlie the capacity to experience emotions and to observe and empathize with the emotions of others. Evidence supporting the hypothesis of a shared capacity comes from several studies. Inhaling odorants that produce a strong feeling of disgust produce similar fMRI patterning to that occurring by observing video clips of people showing facial expression of disgust [102]. Observing and feeling disgust appear to activate the same sites in the anterior insula and anterior cingulate cortex. This finding has been replicated in other laboratories [103]. The shared region appears to be embedded in distinct functional circuits during observing, imagining and experiencing an emotion, which helps to explain why these three processes feel different to the perceiver [104]. The mirroring of emotion appears to play a unifying pro-social role in sharing and synchronizing people's actions and emotions. ACT holds that mirror neurons alone cannot explain perception without the integration provided by schemata. Schemata provide the necessary internal models to mediate neural mirror responses within the action control cycle.

Multiple examples of applications to action observation treatments (AOT) have been trialed and tested for patients with stroke, Parkinson's disease, multiple sclerosis, cerebral palsy, orthopedic trauma and postsurgical patients [105]. Parkinson's disease (PD) alters gait patterns from early stages, and visual-motor training strategies such as action observation (AOT) and motor imagery (MI) that are based on the activity of the mirror neuron system (MNS) can facilitate motor re-learning. A systematic review summarized the evidence of the effectiveness of MNS treatments (AO and MI) to treat gait in patients with PD [105]. The effects of AO and MI were referenced in terms of disease severity, quality of life, balance and gait. Training with AO and MI was found to be effective in improving disease severity, quality of life, balance and gait in patients with PD [106].

### 9. Limitations

This review has focused on the visual system. Yet perception and mental imagery are known to be multimodal experiences across multiple senses. The focus has been on psychological research, and other multidisciplinary perspectives on vividness have not been covered [107]. This review has also covered principally questionnaire-based vividness scores from the VVIQ and similar instruments rather than trial-based vividness scores that are sensitive to momentary variations and are known to yield stronger associations between vividness, brain states and behavior [108]. Thus, the reviewed findings may under-represent the full strength of vividness effects. Another limitation is the key role attributed to three hypothetical kinds of schemata (S, M and P types) when, to date, no study has definitively identified the neural cells and regions that embody these entities. This issue remains a challenge for future investigation. Finally, in spite of its significance, the role of VMI in human imagination and creativity [32,109] has not been included in this review.

### 10. Conclusions

The study of mental imagery has contributed to our basic understanding of vision.

The ACT provides a coherent and feasible organization for perception and mental imagery. The six modules and their interconnections receive empirical support from a wide range of studies.

All six modules of the perception and mental imagery system are influenced by individual differences in vividness.

Real-world applications of ACT show interesting potential to improve human wellbeing in both healthy people and patients.

Mental imagery can be applied in creative ways to make new collective goals and actions for change that are necessary to maximize the future prospects of the planet.

**Funding:** The writing of this review received no external funding.

**Institutional Review Board Statement:** No new data were collected for this review article, and no institutional review was necessary.

**Informed Consent Statement:** Not applicable.

**Data Availability Statement:** Not applicable.

**Acknowledgments:** I acknowledge the contributions of hundreds of investigators whose publications are included in the Reference list for this review.

**Conflicts of Interest:** The author declares no conflict of interest.

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
