# Peer review of "The Action Cycle Theory of Perception and Mental Imagery"

_2411-5150, 2022_

Round 1

Reviewer 1 Report

In this paper, the creator of the widely used vividness imagery questionnaire (VVIQ) proposes an Action Control Theory (ACT) with six components: Schemata, Objects, Actions, Affect, Goals and Others’ Behavior. He then reviews some empirical evidences supporting each component, based on the studies quantifying subjective imagery vividness by the VVIQ questionnaire. The goal to the paper is ambitious: to provide a review about the interactions between the visual system (both perceptual and imaginal) and other systems (emotion, motivation, movement etc), and to examine the impact of subjective imagery vividness on the interactions. However, the paper is not easy to follow, and some of the studies appear to be cherry-picked to support the author’s claims. More specific points follow.

Major points:

1.      The structure of the paper is a bit loose. It starts with a presentation of VVIQ and a defense against some objections that have been raised against its use. While it is understandable that the creator of VVIQ is (rightfully) proud of this questionnaire, it is less clear how this section contributes to the model that the author wishes to develop. Then the six processes of the model should be examined in turn, but this is not really the case. For example, the role of hippocampus in the Schemata section is unclear. 

2.      The reasons for selecting this specific six components are not clear. 

3.      Causal relationships between the components are schematized in the Figure 2. However, the reasons of the arrow direction are unclear. For example, why is the arrow between goals and schemata unidirectional? More importantly, where are perception and mental imagery in the figure?

4.      What is the difference between the sections Action and Others’ behavior, concerning the discussion on mirror neurons? 

5. Line 202. “There appears to be in-built redundancy in the VMI network as evidenced by the reported ability to experience vivid VMI in the absence of significant amounts of cortical tissue.” Actually, as for all other cognitive functions it depends a lot on the specific loci of cortical damage, as well as on their connections. Specifically, lesions restricted to the occipital cortex, such as those that provoked near-complete cortical blindness in the patient mentioned by the author, typically leave visual mental imagery unimpaired. In contrast, lesions or disconnections in the left temporal lobe can impair this ability. See, e.g., DOI: 10.1016/j.cortex.2006.06.004 and 10.1038/s41583-020-0348-5. Potential contributions of the cerebellum or the hippocampus to visual mental imagery are much less well established. Contrary to the author’s statements, even neuroimaging data are consistent with this theoretical landscape: see DOI: 10.1016/j.neubiorev.2020.12.029.

6. Line 216. “vividness correlated with early visual cortex activity“. However, another study found an _inverse_ correlation (DOI: 10.1016/j.cortex.2017.09.014). See also the previous point on the evidence from cortically blind patients with extensive damage of V1 and preserved imagery. See also line 230: Cui et al. (2007)  found the relative visual cortex signal averaged over the visualization window correlates negatively  with the subjective rating of vividness. But in Figure 6A, Amedi et al. (2005) found a trend for positive correlations in V1 (primary visual cortex) with the imagery vividness. The author might want to discuss these discrepancies about the role of V1 activity in his model.

7. The author states that “vivid images are associated with fewer eye movements” (line 234), but offers no explanation of this finding.  If images are built in V1, one would rather think that the peripheral parts of vivid images would be more likely to “capture” foveating eye movements, as it happens in perception.

8. Line 297. It is not clear to me how the author’s framework challenges O’Regan’s theory. Please explain.

9. Section 9. Are there any predictions concerning intrusive imagery in PTSD?

Minor points:

  1. Six domains, six processes or six components? The label should be consistent throughout the MS. 
  2. Line 51 “Although Neisser’s theory appeared flawed” - Why? Please explain.
  3. In the citation starting from line 129, “lessor” should be “lesser”
  4. Line 178. “objectively quantifiable using measurements”: “using” seems out of context here.
  5. Line 179, “Tabi et al. found a significant relationship between VVIQ scores and the volumes of primary visual cortex.” But see DOI: 10.1093/cercor/bhv186 for an inverse result.
  6. There are three Sections 8:  Others’ behavior, Limitations and Conclusions.

Author Response

RESPONSE TO REVIEWER A

Thank you for your helpful and constructive review. I address the points you have raised one by one.

In this paper, the creator of the widely used vividness imagery questionnaire (VVIQ) proposes an Action Control Theory (ACT) with six modules: Schemata, Objects, Actions, Affect, Goals and Others’ Behavior. He then reviews some empirical evidences supporting each module, based on the studies quantifying subjective imagery vividness by the VVIQ questionnaire. The goal to the paper is ambitious: to provide a review about the interactions between the visual system (both perceptual and imaginal) and other systems (emotion, motivation, movement etc), and to examine the impact of subjective imagery vividness on the interactions. However, the paper is not easy to follow, and some of the studies appear to be cherry-picked to support the author’s claims.

This paper is an attempt to present a convincing case in support of the author’s theory of perception and mental imagery. I have not cherry-picked studies although there is a vast orchard of cherries to pick. I have focused on studies which correlated VVIQ scores with neural or cognitive performance. In the revised version I have pointed  out that uncertainties exist and different findings are obtained by different researchers especially with small-N studies that tend not to replicate. I have improved the paper’s clarity and structure. I hope the final version, which is largely revised with improvements provoked by your review, gives a more cogent and balanced impression to the reviewer.

More specific points follow.

Major points:

  1. The structure of the paper is a bit loose. It starts with a presentation of VVIQ and a defense against some objections that have been raised against its use. While it is understandable that the creator of VVIQ is (rightfully) proud of this questionnaire, it is less clear how this section contributes to the model that the author wishes to develop.

Thank you for this point. I have tightened both the argument and the structure.

The foundation of the empirical evidence presented in the article rests on studies that have used the VVIQ. It appears helpful to indicate the credentials of the instrument as a ‘safe’ foundation by summarizing a few critical issues that have been raised. Neglecting to mention issues like validity, subjectivity and social desirability bias would detract from the credibility of the VVIQ as a source of evidence.

To improve the structure, I have moved the three paragraphs about the VVIQ from section 2 into section 1, so section 2 remains solely about the theory. I believe this change is beneficial to the description of the theory.

Then the six processes of the model should be examined in turn, but this is not really the case. For example, the role of hippocampus in the Schemata section is unclear. 

I have tightened the structure to examine the six modules of the theory one-by-one in sections 3-8. I explain in more depth and detail the role of the hippocampal/cerebellar system in the Schemata module (section 3). I describe and reference the review by Parkins [50] (l 196-206).

  1. The reasons for selecting this specific six modules are not clear. 

I have added a sentence (l 114-115) to explain that the six modules are viewed as necessary and sufficient for perception and mental imagery to occur. According to Action Cycle Theory, all six modules are required by the system, which would not function if one or more modules were missing.

  1. Causal relationships between the modules are schematized in the Figure 2. However, the reasons of the arrow direction are unclear. For example, why is the arrow between goals and schemata unidirectional?

I have added more detail to the legend for Figure 2, which states:The majority of interconnections are reciprocal meaning that any module can causally influence, and be causally influenced by, its immediate neighbours. The absence of an arrow going from Schemata to Goals indicates that Schemata do not influence Goals, which are determined by Affect only, i.e., the degree to which  happiness or pleasure can be increased and/or pain can be diminished.”

More importantly, where are perception and mental imagery in the figure?

I have made the answer to this question as clear as possible so that the majority of readers will ‘get it’. As stated under 2 above, “the six modules are viewed as necessary and sufficient for perception and mental imagery to occur.”  To shadow the traditional S-R concept, I have added two large grey arrows to Figure 2 indicating (i) the sensory input entering the system via the Object module and (ii) the output of the system via the Action module. Perception and mental imagery are both produced by the same six modules of the action cycle but in the absence of sensory input in the case of mental imagery. To emphasize this point, the legend of Figure 2 and the text have been revised. At lines 113-116, I state:Action Cycle Theory (ACT) is an enactive theory of perception and mental imagery with six interrelated modules (Figure 2).  Figure 2 summarizes the theory in showing that perception and mental imagery are constituted by the six modules, which are: Object, Schemata, Action, Affect, Goals and Other’s Actions.”

5).

  1. What is the difference between the sections Action and Others’ behavior, concerning the discussion on mirror neurons? 

‘Action’ refers to the perceiver’s/imager’s own actions. ‘Others’ behavior’ refers to the actions of others. I have added a sentence to clarify this point (l 117). Mirror neurons participate whenever Other’s actions are being observed. They are not a necessity for self-generated action unless one is basing an action on a mental image of another actor. (I 488-490).

  1. Line 202. “There appears to be in-built redundancy in the VMI network as evidenced by the reported ability to experience vivid VMI in the absence of significant amounts of cortical tissue.” Actually, as for all other cognitive functions it depends a lot on the specific loci of cortical damage, as well as on their connections. Specifically, lesions restricted to the occipital cortex, such as those that provoked near-complete cortical blindness in the patient mentioned by the author, typically leave visual mental imagery unimpaired. In contrast, lesions or disconnections in the left temporal lobe can impair this ability. See, e.g., DOI: 10.1016/j.cortex.2006.06.004 and 10.1038/s41583-020-0348-5. Potential contributions of the cerebellum or the hippocampus to visual mental imagery are much less well established. Contrary to the author’s statements, even neuroimaging data are consistent with this theoretical landscape: see DOI: 10.1016/j.neubiorev.2020.12.029.

I revised the sentence as follows: “There appears to be in-built redundancy in the VMI network as evidenced by the reported ability to experience vivid VMI in the absence of significant amounts of cortical tissue (depending, of course, on the amount and location of the missing or damaged tissue).”

(l 229-232).

To indicate the current uncertainty about the cortical representation of VMI, I added (l 237-245): “To date, no consensus has yet been established about the cortical systems involved in VMI. One large-scale meta-analysis of 46 fMRI studies [56] revealed that VMI engages fronto-parietal networks and a region in the left fusiform gyrus. Spagna et al. [56] proposed a revised neural model of VMI with fronto-parietal networks initiating, modulating, and maintaining activity in a core temporal network centered on the fusiform imagery node, a high-level visual region in the left fusiform gyrus. The cortical basis of VMI remains a subject of active investigation.”

  1. Line 216. “vividness correlated with early visual cortex activity“. However, another study found an _inverse_ correlation (DOI: 10.1016/j.cortex.2017.09.014). See also the previous point on the evidence from cortically blind patients with extensive damage of V1 and preserved imagery. See also line 230: Cui et al. (2007)  found the relative visual cortex signal averaged over the visualization window correlates negatively  with the subjective rating of vividness. But in Figure 6A, Amedi et al. (2005) found a trend for positive correlations in V1 (primary visual cortex) with the imagery vividness. The author might want to discuss these discrepancies about the role of V1 activity in his model.

I agree that different studies, especially those using small Ns, tend not to be replicated by other small N studies. I added a sentence saying this at the end of the legend for Figure 2: “Note that there were only 8 participants in this study and the p level for the correlation was only .04. Other studies have obtained different results [57, 60].”

However, the findings of Cui et al. are not in conflict with those of Amedi et al. Cui’s 8 subjects showed wide differences in their relative fMRI signal in visual cortex ranging from -0.3 to +0.2 and with a pronounced u-shaped or inverse u-shaped curves over time (0-20 sec). Overall, using a single score for each subject averaged over the window there were 6 of 8 subjects with positive fMRI signal scores. The negative correlation concerns the fMRI signal scores vs vividness scores, where a low vividness score means high vividness. Figure 5c shows that the 5 of 8 subjects with the positive fMRI signal scores also had average VVIQ scores below 2.5 (high vividness). This accords with the finding of Amedi et al (2005) of positive fMRI scores in V1 correlating with vivid imagery.

  1. The author states that “vivid images are associated with fewer eye movements” (line 234), but offers no explanation of this finding.  If images are built in V1, one would rather think that the peripheral parts of vivid images would be more likely to “capture” foveating eye movements, as it happens in perception.

The findings of [58] and [59] are consistent in showing that vivid imagers make fewer eye movements in VMI. It is likely that keeping the eyes still provides the benefit of avoiding new perceptual input. This hypothesis is consistent with Amedi et al’s [60] findings that people who are visually imaging also ‘switch off’ the auditory modality with auditory cortex deactivation negatively correlating with activation in visual cortex and with the score in the subjective vividness of visual imagery questionnaire (VVIQ). I have added text to make this point explicit (i. 273-277).

  1. Line 297. It is not clear to me how the author’s framework challenges O’Regan’s theory. Please explain.

See l. 56.

  1. Section 9. Are there any predictions concerning intrusive imagery in PTSD?

Not in this paper. I deal with PTSD elsewhere (Marks, 2023, in preparation).

Minor points:

  1. Six domains, six processes or six components. The label should be consistent throughout the MS. 

I have aimed for consistency in the revised draft by referring to ‘modules’ rather than to components, domains or processes.

  1. Line 51 “Although Neisser’s theory appeared flawed” - Why? Please explain.

Please see Hampson and Morris (1978) doi.org/10.1016/0010-0277(78)90010-0: Anticipation may well be a necessary but is not a sufficient module of imaging. (3) Neisser's account has severe problems in explaining how images are manipulated and used in cognition. (4) The conscious experience of having an image is seen to differentiate imaging from just knowing — a possibility not admitted by Neisser. (5) Finally, Neisser's view of introspection is criticized on the grounds that the use of real world descriptive terms does not imply a description of real world objects. Introspection, whilst not error free, still occurs.” I agree with these criticisms. I think it would be an unnecessary distraction to mention the reasons why Neisser’s theory appeared flawed but I could add the reasons if need be.

  1. In the citation starting from line 129, “lessor” should be “lesser”

I could not locate this.

  1. Line 178. “objectively quantifiable using measurements”: “using” seems out of context here.

Deleted ‘using’.

  1. Line 179, “Tabi et al. found a significant relationship between VVIQ scores and the volumes of primary visual cortex.” But see DOI: 10.1093/cercor/bhv186 for an inverse result.

My reading of the paper is that the study you cite investigated primary visual cortex (V1) surface area, not volume, and they did not investigate hippocampal volume. However, I have removed any mention of primary visual cortex volume here.

  1. There are three Sections 8:  Others’ behavior, Limitations and Conclusions.

The sections have been renumbered sequentially.

Thank you again for your constructive review, which has led to an improved version of the paper. I hope you agree.

Reviewer 2 Report

I do not have any issues or concerns. I would just ask the author to work a bit on revising parts of the manuscript that are similar to previoulsy published materials. I know that this is difficult because this is a review based on the author's own extensive prior work, a life time career contribution. Nevertheless, I think the effort may pay off producing a crisper paper.

Author Response

RESPONSE TO REVIEWER 2

Thank you for your positive remarks and ratings. I have tried my level best not to repeat myself. 

Reviewer 3 Report

The paper reviews studies done on imagery in the context of vividness quite well. 

Author Response

RESPONSE TO REVIEWER 3

Thank you for your positive comment. 

Round 2

Reviewer 1 Report

I wish to thank the author for taking into consideration all my points. 

The author changed the name of his theory (control became cycle). This should be reflected in the abstract.